# Co-Culture of Cryopreserved Healthy Sertoli Cells with Testicular Tissue of Non-Obstructive Azoospermia (NOA) Patients in Culture Media Containing Follicle-Stimulating Hormone (FSH)/Testosterone Has No Advantage in Germ Cell Maturation

**DOI:** 10.3390/jcm12031073

**Published:** 2023-01-30

**Authors:** O. Sena Aydos, Yunus Yukselten, Tulin Ozkan, Sinan Ozkavukcu, Meltem Tuten Erdogan, Asuman Sunguroglu, Kaan Aydos

**Affiliations:** 1Department of Medical Biology, School of Medicine, Ankara University, Ankara 06230, Turkey; 2Department of Internal Medicine, Section of Infectious Diseases, Yale School of Medicine, New Haven, CT 06520, USA; 3Center for Assisted Reproduction, School of Medicine, Ankara University, Ankara 06230, Turkey; 4Postgraduate Medicine, School of Medicine, University of Dundee, Dundee DD1 4HN, UK; 5Department of Urology, School of Medicine, Ankara University, Ankara 06230, Turkey

**Keywords:** Sertoli cells, cryopreservation, NOA, FSH, testosterone

## Abstract

Different cell culture conditions and techniques have been used to mature spermatogenic cells to increase the success of in vitro fertilization. Sertoli cells (SCs) are essential in maintaining spermatogenesis and FSH stimulation exerts its effect through direct or indirect actions on SCs. The effectiveness of FSH and testosterone added to the co-culture has been demonstrated in other studies to provide microenvironment conditions of the testicular niche and to contribute to the maturation and meiotic progression of spermatogonial stem cells (SSCs). In the present study, we investigated whether co-culture of healthy SCs with the patient’s testicular tissue in the medium supplemented with FSH/testosterone provides an advantage in the differentiation and maturation of germ cells in NOA cases (N = 34). In men with obstructive azoospermia (N = 12), healthy SCs from testicular biopsies were identified and purified, then cryopreserved. The characterization of healthy SCs was done by flow cytometry (FC) and immunohistochemistry using antibodies specific for GATA4 and vimentin. FITC-conjugated annexin V/PI staining and the MTT assay were performed to compare the viability and proliferation of SCs before and after freezing. In annexin V staining, no difference was found in percentages of live and apoptotic SCs, and MTT showed that cryopreservation did not inhibit SC proliferation compared to the pre-freezing state. Then, tissue samples from NOA patients were processed in two separate environments containing FSH/testosterone and FSH/testosterone plus co-culture with thawed healthy SCs for 7 days. FC was used to measure 7th-day levels of specific markers expressed in spermatogonia (VASA), meiotic cells (CREM), and post-meiotic cells (protamine-2 and acrosin). VASA and acrosin basal levels were found to be lower in infertile patients compared to the OA group (8.2% vs. 30.6% and 12.8% vs. 30.5%, respectively; *p* < 0.05). Compared to pre-treatment measurements, on the 7th day in the FSH/testosterone environment, CREM levels increased by 58.8% and acrosin levels increased by 195.5% (*p* < 0.05). Similarly, in medium co-culture with healthy SCs, by day 7, CREM and acrosin levels increased to 92.2% and 204.8%, respectively (*p* < 0.05). Although VASA and protamine levels increased in both groups, they did not reach a significant level. No significant difference was found between the day 7 increase rates of CREM, VASA, acrosin and protamine-2 in either FSH/testosterone-containing medium or in medium additionally co-cultured with healthy SCs (58.8% vs. 92.2%, 120.6% vs. 79.4%, 195.5% vs. 204.8%, and 232.3% vs. 198.4%, respectively; *p* > 0.05). Our results suggest that the presence of the patient’s own SCs for maturation of germ cells in the culture medium supplemented with FSH and testosterone is sufficient, and co-culture with healthy SCs does not have an additional advantage. In addition, the freezing–thawing process would not impair the viability and proliferation of SCs.

## 1. Introduction

In a recent review, the incidence of non-obstructive azoospermia (NOA) among infertile men is stated as approximately 10–15% [1]. Due to the presence of small foci of spermatogenesis in the testis, it is possible for these men to be biological fathers by using sperm cells retrieved from these areas. Hereby, in 54% of NOA cases, spermatozoa can be obtained with microdissection testicular sperm extraction (microTESE) [2]. According to a recent meta-analysis, retrieved sperms resulted in live birth rates in up to 28% of intracytoplasmic sperm injection (ICSI) cycles [3]. Recently, the use of round spermatids that could not progress to the mature spermatozoa stage but have completed meiosis has also been suggested as a last resort for a group of NOA cases to have their own genetic offspring [4]. However, the live birth rate after round spermatid injection (ROSI) was considerably lower than after ICSI using testicular spermatozoa because round cells have not completed the advanced differentiation stage. Therefore, there is still a need for an effective treatment approach for TESE-failed NOA cases.

Although some progress has been made in deriving male germ cells from embryonic stem cells (ESCs) and induced pluripotent stem cells (iPSCs), an optimized approach to obtaining functional spermatozoa from human spermatogonia has not been achieved at present, aside from ethical issues and tumor-forming risks. In vitro maturation of early-stage germ cells has been a promising treatment option in azoospermia cases where mature sperm cells cannot be obtained from testis. In preliminary studies, it was shown that maturation up to mature spermatozoa can be induced by co-culture of round spermatids on Vero cell monolayers [5]. Similar findings have also been reported by Tanaka et al., where maturation of primary spermatocytes into haploid spermatids could be successfully achieved through in vitro co-culture with Vero cells [6]. Subsequently, Yang et al. used retinoic acid (RA) and stem cell factor (SCF) to culture human SSCs of cryptorchid patients and developed haploid spermatids possessing fertilization and development capacity [7]. More recent researches are being conducted on the development of many different culture systems such as human amnion mesenchymal stem cells [8], amniotic fluid-derived exosomes [9], in vitro reprogramming of fibroblasts to human induced Sertoli-like cells (hiSCs) [10], 3D culture [11,12,13], isolated cell culture with growth factor supplementation [14], and organotypic culture [15]. These studies expanded our knowledge of the interactions between somatic cells, germ cells, and the extracellular matrix, as well as the impact of regulatory pathways in spermatogenesis.

The Sertoli–germ cell interactions are pivotal in maintaining spermatogenesis by providing structural and functional support to germ cells. The niche provided by Sertoli cells (SCs) is known to be key in spermatogenesis and morphologically demonstrated Sertoli cell defects have been previously documented in NOA cases [16]. SCs regulate spermatogenetic activity by secreting various molecules such as glial cell line-derived neurotrophic factor (GDNF), SCF, fibroblast growth factor 2 (FGF2), and bone morphogenic protein 4 (BMP4) [17]. Recently, abundant miRNAs in SC-derived extracellular vesicles were also shown to provide intercellular communication between SCs and SSCs [18].

Pituitary gonadotropins have a crucial importance in the regulation of SC function [19]. FSH stimulation exerts its effect through direct or indirect actions on SCs by up-regulating specific proteins related to spermatogenesis and the last stage of spermiation [20,21]. Besides, FSH has been shown to have an effective role in the survival and proliferation of early-stage germ cells [22]. In culture medium containing FSH, type A spermatogonia can progress up to the pachytene spermatocyte stage within three weeks [23]. On the other hand, the completion of meiosis is androgen-dependent and is regulated via AR on SCs [24]. In response to testosterone, SCs provide completion of meiosis by modulating post-translational events in spermatocytes. Tanaka et al. reported that when early round spermatids were isolated and co-cultured with FSH and testosterone on Vero cell monolayers, in vitro spermiogenesis rates were increased [25].

It is argued that the contributions of FSH and testosterone in the maintenance of normal spermatogenesis are managed through independent pathways [26]. Therefore, it has been shown that meiosis of diploid germ cells and spermiogenesis can be resumed by adding FSH and testosterone together in the medium co-cultured with SCs obtained from testes with maturation arrest [22]. The use of FSH and testosterone for the differentiation and maturation of germ cells has also been suggested in other in vitro studies [27,28]. However, it is not clear whether native SCs are sufficient for FSH and testosterone added to the culture medium to be effective in the induction of spermatogenesis, and whether co-culture with healthy SCs will increase this activity. The aim of this study is to elucidate the necessity of adding healthy SCs to the medium used to restore spermatogenesis. For this purpose, we investigated whether co-culturing healthy donor SCs with the patient’s testicular tissue suspension containing its own SCs in the medium supplemented with FSH/testosterone has an advantage in the differentiation and maturation of germ cells in NOA cases.

## 2. Materials and Methods

Two hundred and twenty-six infertile patients with clinical diagnosis of NOA and 12 men with obstructive azoospermia (OA) as SC donors were included in our study. The study was approved by the Ankara University Ethics Committee. The patients gave informed consent to use testicular tissues for the experiment. Medical histories were obtained for all patients and a detailed urological examination was performed. Ejaculate samples taken with an interval of at least 2 weeks were evaluated according to the WHO criteria [29]. The absence of dead or alive sperm cells in the pellet after high-speed centrifugation was defined as azoospermia. Hormone measurements, including follicle stimulating hormone (FSH) (reference range 1.0–10.5 IU/mL) and total testosterone (reference range 2.0–12 ng/mL), were made by chemiluminescent immunoassay (Beckman Coulter UniCel DxI 800 System, USA). Testicular volumes were measured by scrotal ultrasonography. The presence of obstruction in the distal ejaculatory ducts was investigated by transrectal ultrasonography. Cases with a medical history of undescended testis, orchids, Y-chromosome deletion, hypogonadotropic hypogonadism, testicular trauma, malignancy, chemotherapy, or radiotherapy were excluded from the study. The histopathological structure of the testicular tissue was determined by examining the biopsy samples taken during scrotal exploration as described previously [30]. A diagnosis of NOA was made based on clinical examination and scrotal exploration findings, as well as testicular biopsy results. Patients with previous vasectomy or bilateral vas agenesis, normal hormone levels, and normal testicular biopsy were accepted as OA. Out of 226 cases, only 34 maturation arrest cases who met these criteria were included in the study, and those with SCO and hypospermatogenesis were not accepted. Maturation arrest was characterized by complete arrest of spermatogenetic maturation up to the spermatocytes level [31]. Tissue samples for use in the study were surgically taken during microTESE intervention as previously described [32].

### 2.1. Study Design

At the beginning of the study, to demonstrate whether the freezing–thawing process would cause damage, Sertoli cells isolated from OA cases were first characterized and subsequently frozen. After the thawing process, the viability and proliferation rate measurements were determined and compared with the baseline values before they were used in co-culture. Then, testicular tissue suspensions of NOA cases with maturation arrest were prepared in 2 separate media, one containing only FSH/testosterone (Group 1) and the other co-cultured with frozen-thawed healthy SCs in addition to FSH/testosterone (Group 2). In both groups, pre-treatment expression levels of CREM, VASA, acrosin, and PRM2 in testicular tissue extracts were compared by flow cytometry with those of the seventh day following their culture.

#### 2.1.1. Healthy Sertoli Cell Isolation and Culture

Isolation and culture of Sertoli cells were done by modification of previously described methods [33,34]. In order to obtain healthy SCs from subjects with OA, testicular tissue samples taken during the TESE procedure were homogenized by cutting with the help of scalpel and micro scissors, then suspended and washed up three times with phosphate-buffered saline (PBS; Sigma, USA) containing 1% penicillin/streptomycin (Lonza, cat # BE17-516F). The minced pieces were suspended in DMEM, which contained an enzyme mixture consisting of 50 μL collagenase type IV (Invitrogen, cat # 17104-019) and 5 μL hyaluronidase (Vitrolife, cat # 10017), followed by incubation for 45 min at 37 °C.

Then, for the inactivation of proteinase enzymes, enzyme inhibitor fetal bovine serum (FBS) (Biowest, cat # S1810-500) was added and the particles were left to settle down. Subsequently, the supernatant was transferred to another vial and centrifuged at 800× *g* for 5 min. The pellet was resuspended in DMEM/F12 medium (Lonza, cat # BE12-719F) containing 10% FBS and 1% penicillin-streptomycin (Lonza, cat # DE17-602E). The suspension was then incubated for 48 h at 37 °C in 5% CO_2_ condition (Sanyo MCO-20AIC). At the end of the incubation, it was washed with PBS containing penicillin/streptomycin and kept in DMEM/F12 medium. Flasks containing SCs were preserved at 37 °C in 5% CO_2_. The medium was changed every two days and any non-adherent, dead cells and cell debris were removed. Passages were performed when the culture flask was covered with 70% cells. For this, the upper layer medium was discarded and the cells were washed with sterile phosphate buffered saline (PBS) (Ca^++^- and Mg^++^-free) (Gerbu, cat # 210307). The PBS was then removed and, after adding 1 mL Trypsin/EDTA (Gibco, cat # R-001-100), the cells were incubated for 5 min at 37 °C. The cell suspension was centrifuged at 1200 rpm for 3 min. After centrifugation, the supernatant portion was discarded. At the end of passage 3, 10% DMSO (Applichem, cat # 67-68-5+) DMEM/F12 freezing medium was added to cells for cryopreservation. Cryopreservation was performed by modifying a previously described method [35]. The freezing medium was incubated for 15 min so that the DMSO could be absorbed from the cell membrane. Cells in suspension were stained with trypan blue and counted with a hemocytometer. Cells to be frozen after being set to 2 to 4 × 10^6^/mL in each vial were first kept at −20 °C for 1 h and then at −80 °C for 24 h in the freezing container, and then removed to tanks containing liquid nitrogen for long-term storage. Thawing was done in a water bath at 37 °C by shaking it continuously until no ice crystals remained and after centrifugation, the cells were suspended in the medium.

#### 2.1.2. Characterization of Sertoli Cells

##### Flow Cytometry

Fixation, permeabilization, and blocking steps were applied consecutively according to a previously described protocol [36]. Briefly, Trypsin/EDTA mixture was used to separate the adherent SCs from the flask surface. Trypsinized cells were centrifuged at 5000 rpm for 5 min after PBS was added. Following re-washing, 4% paraformaldehyde was added to the pellet for fixation and after 20 min incubation at room temperature in the dark, washing was repeated twice. For permeabilization, 0.1% triton X-100 (GERBU Biotechnik, GmbH) was added onto the pellet and incubated at room temperature for 15 min. After incubation, the pellet was washed twice with PBS. In the blocking stage, cells were incubated in PBS containing 3% BSA (Sigma-Aldrich, Darmstadt, Germany) for 30 min at room temperature, then centrifuged at 5000 rpm for 5 min and the supernatant was removed. The pellet was dissolved in blocking buffer and divided into 4 tubes. The first tube was reserved for unstained, the second tube for isotype, the third tube for vimentin (Vimentin D21H3 XP^®^ Rabbit mAb, cat # 5741S), and the fourth tube for GATA4 (Anti-GATA4 antibody, ABCAM, cat # ab134057). Primary antibodies were added as corresponded and the tubes were incubated at room temperature in the dark for 30–45 min, followed by washing twice with PBS. Secondary antibody (FITC) (FITC-goat anti-rabbit secondary antibody, Invitrogen, cat # 656111) as an isotype was added to the second, third, and fourth tubes. After incubating for 30 min at room temperature, the cells were washed twice. The pellet was resuspended in PBS and analyzed in the ACCURI C6 flow cytometer device (Accuri C6, Becton-Dickinson and Company, Franklin Lakes, NJ, USA). The results were recorded as percentages. AC16 cell lines were used as positive controls for GATA4 to compare the flow cytometry results.

##### Immunofluorescence

Expressions of Vimentin and GATA4 proteins in SCs were detected at the protein level by immunofluorescence staining with specific antibodies as described previously [37]. One million SCs per well were seeded into a 24-well plate where glass coverslips were placed at the bottom of each well. After 24 h, the medium on the adhering cells at the bottom of the well was removed and the cells were washed twice with PBS. At this stage, 3.5% paraformaldehyde (PFA) was added to each well and after 20 min, the cells were washed again. Parafilm was laid in a petri dish and 10 μL of vimentin and GATA4 antibodies were placed on it. Subsequently, the coverslips with the attached cells, which were removed with the aid of a forceps and washed with PBS, were covered with antibodies. This was incubated for 45 min to allow binding of the primary antibodies. Later, the coverslips were treated with secondary antibodies prepared by similar procedures and incubated for 45 min. Then, the coverslips were lifted with forceps, washed with PBS and covered with previously prepared slides. The edges of these slides were fixed and kept in the dark to dry. Slides were examined under a fluorescence microscope (Olympus BX51 Corporation, Tokyo, Japan).

#### 2.1.3. Studying the Viability (Apoptosis) and Proliferation of SCs

##### FITC-Conjugated Annexin V/PI Staining

To understand whether cryopreservation affects the viability of frozen-thawed SCs, total cell apoptosis was evaluated by annexin V/PI staining assay before and after freezing. The binding rates of annexin V, a marker with high affinity for the early apoptotic surface marker phosphatidylserine, were determined by flow cytometry. Flow cytometry was performed using a FITC Annexin-V Apoptosis Detection Kit according to the described protocol (BD Pharmingen, cat # 556547). Briefly, 10^6^ cells in 1 mL of medium were transferred to Eppendorf tubes and centrifuged at 2500 rpm. After repeating the washing with PBS, 200 μL of 1X Binding Buffer was added to the pellet and labeled with 2.5 μL annexin V and 2.5 μL propidium iodide (PI) in a separate tube. After incubating for 15 min in the dark, the pellet was mixed with 1X Binding Buffer and then flow cytometry analysis was performed with the AccuriTM C6 device. For each patient, the population was selected according to their own unstained cell samples and gate was set according to their own isotypes.

##### MTT Assay

MTT [3-(4,5-dimethylthiazol-2-yl)-2,5-diphenyltetrazolium bromide] analysis was performed to determine if there was a change in the proliferation rate of cryopreserved SCs after thawing. In each well of a 96-well plate, 100 μL of the medium mixture, adjusted to contain 1 × 10^4^ cells, was homogeneously placed. After the 24th, 48th, and 72nd hours, the medium was removed and 100 μL of MTT solution mixture was added to each well (Sigma-Aldrich, Darmstadt, Germany). The plate was kept at 37 °C with 5% CO_2_ for 2 h. Then, the MTT mixture in the wells was discarded and 100 μL of MTT solvent solution was added to each well. Reduction of the yellow tetrazolium salt MTT to purple formazan crystals by metabolically active viable cells was quantified by measuring absorbance at 550–690 nanometers using a multi-well spectrophotometer (BioTek Instruments, Inc., VT, USA). Mean values were calculated in each case after the measurements were repeated 6 times. The results were obtained in the Gen5 software computer program and transferred to an Excel table.

### 2.2. Preparation of Testicular Tissue for Co-Culture

Testicular tissue samples were taken from NOA and OA patients during TESE as described previously [32]. Briefly, under sedation anesthesia supported by local anesthesia, a scrotal midline incision was made and the most suitable testis was explored first. Dilated and opaque tubules were distinguished under the magnification of the operating microscope, excised, and placed in sperm-washing medium (SWM) (G-IVF, Vitrolife, Gothenburg, Sweden). Germ cells were identified first in the operating room and then in the embryology laboratory for detailed examination. In the case of no mature spermatozoa, the same procedures were applied to the other testis. The tissue suspension to be used in the co-culture studies was then washed three times with PBS containing 1% penicillin/streptomycin. Then, it was transferred to a petri dish and homogenized by cutting with the help of scalpel and micro scissors. One milliliter of collagenase type IV was added and mixed at 37 °C for 45 min. DMEM-F12 medium containing 1 ml 10% FBS enzyme inhibitor and 1% penicillin/streptomycin was added. After the macro particles were allowed to settle, the supernatant was centrifuged at 800× *g* for 5 min. Following the washing steps, the suspension was prepared in DMEM-F12 medium containing 10% FBS and 1% penicillin/streptomycin.

### 2.3. Co-Culture Studies

Testicular tissue suspensions were processed in two separate environments containing FSH/testosterone and FSH/testosterone plus frozen-thawed healthy SCs. The samples were divided into three parts. The first portion was left to be fixed for preliminary examination. The second portion was cultured on a 24-well plate in basal medium (Group 1). The last portion of the cells was added onto healthy SCs on a 24-well plate that had been incubated for 10 days to become confluent (Group 2). All samples were incubated for 48 h at 37 °C in 5% CO_2_. At the end of the 48 h, 1 mL of medium containing FSH 2.35 IU/mL (100 ng/mL) (Follitropin beta, Puregon; NV Organon, Oss, the Netherlands) [38] and testosterone (100 nM) [39] were added to each well. Subsequently, the incubation was continued for 7 days under the same conditions.

### 2.4. Measuring Germ Cell Markers by Flow Cytometry

The expression rates of CREM, VASA, acrosin, and PRM2 in the testicular cell suspension prepared in medium with or without healthy SCs were measured in both basal and post-culture samples by flow cytometry as described above. Primary antibodies were obtained from A, B, C, and D for CREM, VASA, acrosin, and PRM2, respectively. As conjugated secondary antibodies, E was used for CREM and VASA, and F for acrosin and PRM2.

### 2.5. Statistical Analysis

In calculating the sample size of the study, the power for each variable was determined as at least 80% and the 1st type error was 5%. Power analysis of the data was done by using the G*Power statistical power analysis program (version 3.1.9.7 Franz Faul, Universitat Kiel, Germany). In comparison with the OA group, the power of the study was found to be 0.80. Descriptive statistics for continuous variables in the study were expressed as mean, standard deviation (SD), and minimum and maximum, whereas for categorical variables, as numbers and percentages. To describe the sample size median, maximum and minimum values were also given. Mann-Whitney-U was used to compare the measurements between the patient-OA groups. A paired t-test was used to compare before and after measurements. Statistical significance level (α) was taken as 5% and the SPSS (IBM SPSS for Windows, ver.24) statistics package program was used for analysis.

## 3. Results

Clinical characteristics of the OA and NOA cases are presented in Table 1. Only testicular volumes and serum FSH levels were significantly different between the groups. No spermatozoa could be detected in any of the NOA patients during microTESE.

### 3.1. Viability of SCs before Freezing and Post-Thawing

According to the flow cytometric results of the annexin V/PI staining assay, the percentages of dead cells (necrotic, early, and late apoptotic cells) before freezing and after thawing were 16.1% and 15.2%, respectively (Figure 1A–C). There was no significant relationship between pre- and post-freezing values (*p* > 0.05). Our findings indicate that the freezing–thawing process does not activate apoptotic mechanisms in SCs, so the damage to SCs in this respect was negligible during co-culture study.

### 3.2. Proliferation Rates of SCs before Freezing and Post-Thawing

The impact of the freezing–thawing process on the metabolic activity of SCs as an indicator of cell proliferation was evaluated by MTT assay. Although the proliferation of the cultured cells tended to increase at 48 and 72 h from the onset time, there was no statistically significant difference between the proliferation rates of SCs at 0, 24, 48, and 72 h before and after freezing (*p* > 0.05) (Figure 2). Although the low number of cells due to difficulties in obtaining and culturing primary Sertoli cells may have been the reason for the low absorbance values, these data show that our freezing and thawing protocol and culture conditions do not cause any cell loss that will affect the study results.

### 3.3. Characterization of SCs

Inverted microscope image of cultured SCs is shown in Figure 1H. For the characterization of Sertoli cells (cell culture passage 3) in TESE material, previously established markers GATA4 and Vimentin were used [40]. After SCs were isolated from testicular tissue samples of two OA cases, they were stained with antibodies specific to Vimentin and GATA4 proteins. The subsequent flow cytometry examination revealed very similar results: 96.9% and 97.3% of the cells were Vimentin+, respectively, and 90.9% and 92.1% of the cells were GATA4+, respectively, confirming the SC identity of most cells (Figure 1D,E).

Expression of GATA4 and Vimentin proteins in SCs were shown in Figure 1I,J, respectively. Fluorescence microscopy showed that more than 90% of the cells expressed proteins specific to SCs. Also, we tested negative (isotype secondary antibody) and positive controls (AC16 cell line) for GATA4 (Figure 1F,G). These molecular results indicate that the isolated cells are compatible with the SC population as they displayed the corresponding specific markers.

### 3.4. Co-Culture Results of SCs with Testicular Tissue Suspension

Flow cytometry examination showed that mean expression levels of CREM, VASA, acrosin, and PRM2 in testicular tissue extracts of OA cases with normal spermatogenesis were 18.3%, 30.7%, 30.6%, and 16.7%, respectively. However, when the same measurements were compared with those of NOA cases not yet cultured with donor SCs, VASA and acrosin levels were significantly lower than those of OA (*p* < 0.05) (Figure 3).

On the 7th day of co-culture of NOA patient-derived cells with donor SCs, expression changes of CREM, VASA, acrosin, and PRM2 were detected relative to baseline values by flow cytometry in Group 1 and Group 2. In both groups, the percentages of increase in CREM and acrosin levels compared to pre-treatment values reached a significant level. While CREM and acrosin in Group 1 increased by 58.8% and 195.6%, respectively, these rates were 92.2% and 204.8% in Group 2. However, when the increase percentages in both groups were compared with each other, there was no significant difference between them (*p* > 0.05) (Figure 4).

## 4. Discussion

In our previous study, we demonstrated that impaired spermatogenesis leading to infertility in NOA patients may be associated with the altered expressions of some SC-derived cytokines. Subsequently, adding FSH to the culture medium restored their levels to normal, revealing that it could have a positive influence on male infertility [36]. However, whether the stimulating effect of FSH supplementation on native Sertoli cells would be beneficial in restoring spermatogenesis was not investigated. Therefore, in the present study, we first aimed to evaluate the stimulating effect of SCs on germ cell maturation in cultures containing FSH and testosterone when used for therapeutic purposes. Besides, we investigated whether co-culturing the testicular tissue suspension containing the patient’s own SCs with healthy donor SCs offered an advantage. However, in order for healthy SCs to be readily available for use when needed, they should be frozen for storage beforehand. Hence, in our experimentally designed study model, the need for cryopreservation of SCs to synchronize the co-culture conditions was crucial. Freezing of biological cells can cause oxidative stress due to excessive ROS accumulation, which can cause peroxidation in lipids, oxidization in proteins, and damage to nucleic acids [41]. Also, the mitochondrial membrane potential was reported to be altered in thawed cells following cryopreservation [42]. Therefore, cryopreservation may cause functional and structural damage in thawed cells by altering cell functions and genome stability. In the present study, to examine whether cryopreservation causes an alteration in the viability and function of SCs, we evaluated the proliferation rates of the cells with the MTT assay and the rates of apoptosis by measuring annexin V expression levels before freezing and after thawing. Labeled annexin V binding is an important indicator in the evaluation of cell viability, as it indicates the presence of phosphatidylserine (PS) exposed on the cell’s outer membrane during apoptosis [43]. We observed that the average viable cell percentages remained 82.0% and 82.6% in both groups, respectively. The results indicated that the freezing–thawing process did not increase the apoptosis rate of SCs and did not change their viability, so the use of cryopreservation of SCs in cultures was a reliable method. Indeed, Baert et al. showed that SOX9-expressing SCs in testicular tissues cryopreserved for one month retained similar dynamics to fresh controls [44]. Likewise, Chui et al. showed that SCs isolated from adult human testes maintained their phenotypic characteristics and functionality following in vitro expansion and cryopreservation [45]. Here, optimal culture conditions may have been effective in enabling the re-synthesis of mitotic and meiotic cell components, and thus in the continuity of cell division and functions. As shown earlier, the FSH we add to the culture medium has an effect on regulating the proliferation of SCs [46].

When basal expression levels of germ cell differentiation markers in testicular tissue extracts were compared before co-culture with donor SCs, VASA expression was found to be significantly lower in NOA cases than in OA cases. VASA is a marker specific to germ cell lineage and is expressed mostly in spermatocytes in the testicles. VASA staining was not observed in spermatids and is therefore considered a specific marker for pre-meiotic early-stage germ cells [47]. Since it has a determinative role in germ cell formation, its low expression in our study explains the impaired spermatogenesis in NOA cases. Acrosin expression was also significantly decreased in our NOA patients. Acrosin is a serine protease localized in the sperm acrosome. Acrosin is used as a marker of advanced germ cell maturation and its expression was detected in pachytene spermatocytes and round spermatids [48]. Since the acrosome begins to form immediately after the completion of meiosis, acrosin synthesis occurs with the initiation of spermiogenesis [49,50]. Considering that the development of the acrosome and the flagellum is a structural change specific to late-stage elongated spermatids, the low acrosin which we observed in our study indicates that the cells in the culture medium had not yet reached the Golgi phase of spermiogenesis. Indeed, it has been reported that in humans, proacrosomal gene transcription begins to become evident in the first haploid round spermatids [51]. Immunochemical studies have also shown that acrosin appears first in early round spermatids and becomes more prominent in elongating spermatids during spermiogenesis [52].

On the other hand, we found that pre-treatment CREM and PRM2 expressions in NOA patients were comparable to those in OA patients. CREM is a transcription factor belonging to the cAMP-mediated signal transduction pathway and plays a major role in regulating the expression of many genes involved in spermatogenesis [53]. Following their expression in haploid round spermatids, the expected structural changes in spermiogenesis begin. Only antagonist forms of the CREM family are found in pre-meiotic germ cells and early prophase spermatocytes. In pachytene spermatocytes, transcripts of activating CREM forms begin to accumulate. However, intensive production is seen in haploid-stage spermatids [54,55]. Elimination of the transcription factor CREM in mice has been shown to prevent the differentiation of spermatids by decreasing post-meiotic gene expression [56]. Also, in azoospermic men with predominant round spermatid maturation arrest, CREM expression is significantly reduced or undetectable. CREM-negative germ cells cannot undergo further maturation [57]. Although we found CREM expression in NOA cases to be similar to that in OA cases, since the tail development in spermatozoa is regulated by multifactorial mechanisms, normal CREM expression alone in our cases with maturation arrest does not require the completion of full germ cell maturation. In fact, CREM mRNA expression can be seen in pachytene spermatocytes and spermatids. Likewise, CREM proteins can be detected in early-stage spermatids [57]. Besides, it is known that CREM expression has a functional relationship with the genes encoding the transition protein TP1 [58]. TP1 is responsible for histone-protamine exchange in the sperm nucleus and can be detected intensively in advanced-stage spermatids. The fact that CREM induces TP1-mediated protaminization, as demonstrated in previous studies, explains the observed expression of PRM2 in our study group together with the presence of CREM [59].

When testicular tissues were cultured in a medium containing FSH and testosterone for seven days in order to induce maturation of germ cells in vitro, all of the spermatogenesis markers we studied showed an increase, but only VASA and acrosin reached a significant level. Similarly, Solomon et al. achieved the induction of proliferation of VASA-expressing pre-meiotic cells and the development of acrosin-expressing post-meiotic cells in vitro from spermatogonial cells obtained from cyclophosphamide-treated mice testis [28]. However, while an acrosin increase was seen only in testosterone-supplemented cultures, the addition of FSH antagonized this effect. Such an effect of FSH, contrary to our findings, may be due to changes that chemotherapy may have made in the cytoskeleton or metabolism of SCs [60]. Indeed, cyclophosphamide has been shown to reduce FSHR expression via mRNA, as demonstrated in ovarian tissue granulosa cells [61]. Therefore, since there was no interaction that would lead to changes in FSHR functions in our study, it can be claimed that FSH has a stimulating effect on acrosin through healthy SCs.

In our study, FSH and testosterone were added to the culture medium to mimic the normal physiology of the pituitary–testis axis for SCs to maintain their normal functions. It has been previously demonstrated that FSH has a stimulating effect on germ cell maturation via FSH receptors on SCs [62]. In isolated SC cultures, different FSH preparations were shown to be effective in the regulation of specific proteins associated with spermatogenic cell migration and in the modulation of proteins involved in the final stage of spermiation [21]. Likewise, testosterone plays a decisive role in germ cell survival and spermiation by acting cooperatively with FSH via AR on the SCs [63]. Recently, co-culture of SSCs with SCs and FSH treatment was shown to promote in vitro self-renewal and proliferation of germline cells [64]. Xenotransplantation of these cells led to the repopulation of spermatogenic cells in the recipient seminiferous tubules. Other studies have also verified that spermatogenesis can be promoted up to round spermatid production in testicular tissue extracts supplemented with FSH and testosterone [22,65]. Moreover, in SCs co-culture studies where recombinant FSH and testosterone supplementation was performed, round spermatids were differentiated to elongating spermatids and in vitro fertilization was achieved in mice [66]. However, even when hormone supplementation was not performed, spermatogenesis from testicular SSCs to fertility-competent sperm formation could be induced in organ cultures using different techniques [67,68]. Nevertheless, the increase in basal germ cell marker expression in testicular tissue culture, although not at a significant level, showed that the hormonal supplementation we performed in our study has an effective role in the maintenance of cellular functions.

Interestingly, in our study, no significant difference in cell differentiation was found between cultures with and without SC supplementation alongside existing native SCs. SCs constitute the most important component of the niche that supports spermatogenesis in the testicles [69]. In testicular cultures, SCs play a paracrine and autocrine role in supporting germ cells by creating both a source and a target for growth regulatory factors [70]. Establishment of spermatogenesis under in vitro conditions has not been fully achieved due to the complex interactions of germ cells and SCs, as well as being dependent on secreted growth factors and cytokines. However, Sertoli cell-derived factors play a major role in the proliferation and self-renewal of SSCs. For this reason, in many studies, a co-culture system has been used in order to benefit from SCs in maintaining and differentiating germ cells with optimum efficiency [71,72]. In early experimental studies, although round spermatids were formed with the achievement of meiosis following the co-culture of germ cells with SCs, spermiogenesis could not be advanced to a further stage [73]. Therefore, it is an acceptable result that donor SCs do not have an additional effect on germ cell maturation compared to existing native SCs in the hormone-supported basal culture conditions we applied in our study.

Several methods have been described for the use of SCs in co-culture studies. For this purpose, immortalized Sertoli cell lines that retain their specific characteristics were established [74,75]. As another alternative, SC feeders were efficiently used for the maintenance and proliferation of germ cells [76]. Nasiri et al. tried four different feeders, including SIM mouse embryo-derived thioguanine and ouabain resistant (STO), mouse embryonic fibroblast, bovine SCs, and on a laminin-coated plate for short-term culturing of testicular germ cells, and STO was found to be the most suitable [77]. As an alternative, we investigated whether co-culturing testicular tissue extracts of NOA cases with healthy SCs would provide an additional benefit in achieving germ cell maturation. As a result, we found that with the addition of healthy SCs to the basal culture medium in testicular cell suspension from NOA patients, the increase in germ cell maturation markers CREM, acrosin, VASA, and PRM2 was similar to that in basal culture. Indeed, although Sertoli cells support the bioactivity of SSCs with the numerous factors and cytokines they secrete, it has been shown that excessive presence of these factors in the medium will also limit the spermatogenesis capacity of SSCs in the testicular niche [78]. The authors emphasized that it is important not to exceed the threshold values at which Sertoli cell derivatives can show maximum activity. Therefore, as we showed in our study, the culture medium containing FSH and testosterone together with their native Sertoli cells may be sufficient for the maturation of the germ cells in the NOA testis to a certain level, and that the addition of healthy SCs does not provide an advantage. Another explanation is that native SCs may have sufficient functional efficiency. In pre-pubertal and transition age children, isolated tubulopathy and SC dysfunction have been identified [79]. In these cases, AMH does not respond to FSH stimulation. However, in men with central hypogonadism, FSH stimulation results in an increase in AMH levels [80]. Therefore, AMH measurement may identify the presence of SC dysfunction, responsible for primary testicular tubulopathy and NOA. In our study, the increase in acrosin and VASA with FSH stimulation in the culture ruled out any SC dysfunction. If SC dysfunction was previously diagnosed, then it would be meaningful to add healthy SCs to the culture medium.

## 5. Conclusions

In conclusion, co-culture medium enriched with FSH and testosterone is an effective treatment for germ cell maturation in testicular tissue extracts. For in vitro maturation of testicular germ cells in cases without SC dysfunction, the presence of the patient’s own SCs in the culture is sufficient and adding healthy SCs does not have an advantage. Thus, it seems that native SCs may be sufficient when complex culture conditions are used for maturation of testicular germ cells in NOA cases. However, evaluating spermatogenetic differentiation in longer culture periods, expanding the study by investigating other markers specific to germ cell maturation, and including cases with documented Sertoli cell dysfunction would yield more efficient results.

## Figures and Tables

**Figure 1 jcm-12-01073-f001:**
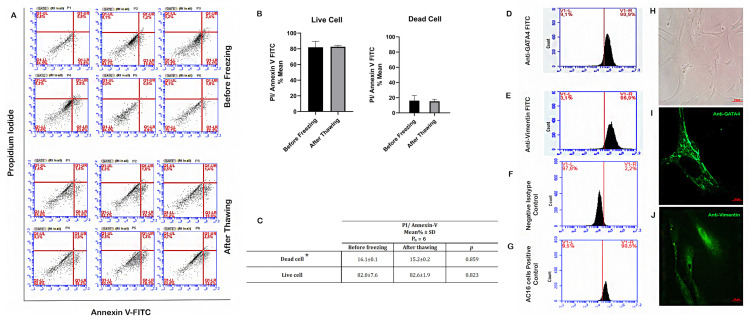
Determination of SC viability by flow cytometry before and after freezing. Flow cytometry plot for annexin V staining of SCs in representative cases. (**A**–**C**) Before freezing and after thawing. (**D**) Flow cytometry plot for GATA4 and (**E**) Vimentin staining of SCs. (**F**,**G**) Flow cytometry plot for negative isotype control and AC16 cells positive control staining for GATA4. (**H**) Inverted light microscope image of primary SCs (100×). (**I**) Fluorescence microscope image of SCs (1000×), green: anti-GATA4. (**J**) Fluorescence microscope image of SCs (1000×), green: anti-Vimentin. * Dead cell: (necrotic + early apoptotic and late apoptotic cells). Pn: number of the patients.

**Figure 2 jcm-12-01073-f002:**
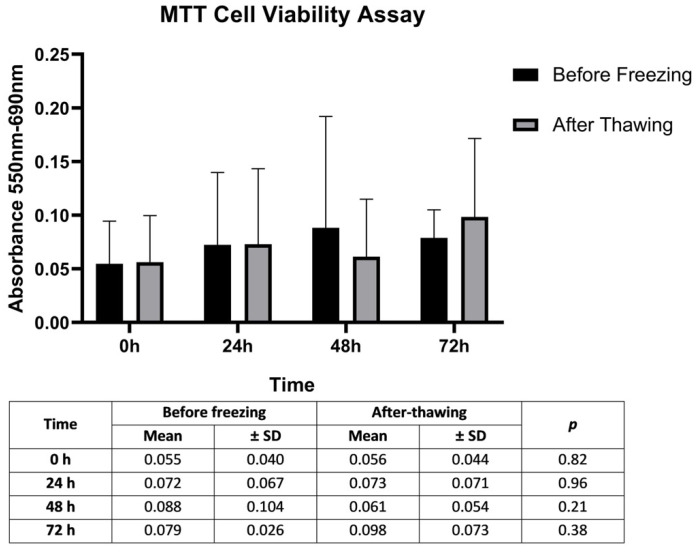
Determination of the cytotoxic activity of the freezing–thawing process by calculating mean proliferation rates of SCs. Significance levels of SCs mean proliferation rates in 12 OA cases at 0, 24, 48, and 72 h according to paired t-test results using the MTT assay. p: comparison of before freezing and after thawing values. Statistical significance level = 0.05.

**Figure 3 jcm-12-01073-f003:**
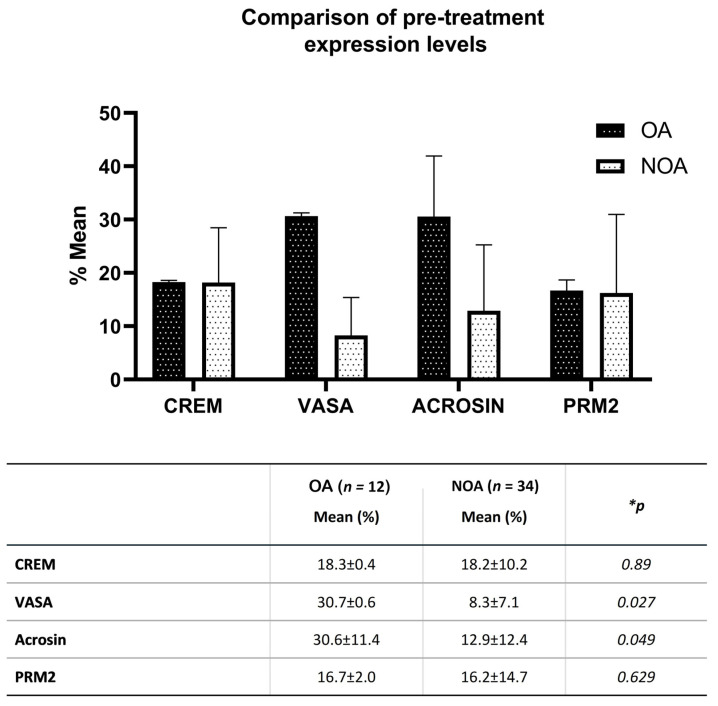
Comparison of pre-treatment expression percentage levels of CREM, VASA, acrosin, and PRM2 in testicular tissue extracts of OA and NOA cases by flow cytometry. * Significance levels according to the Mann-Whitney U Test.

**Figure 4 jcm-12-01073-f004:**
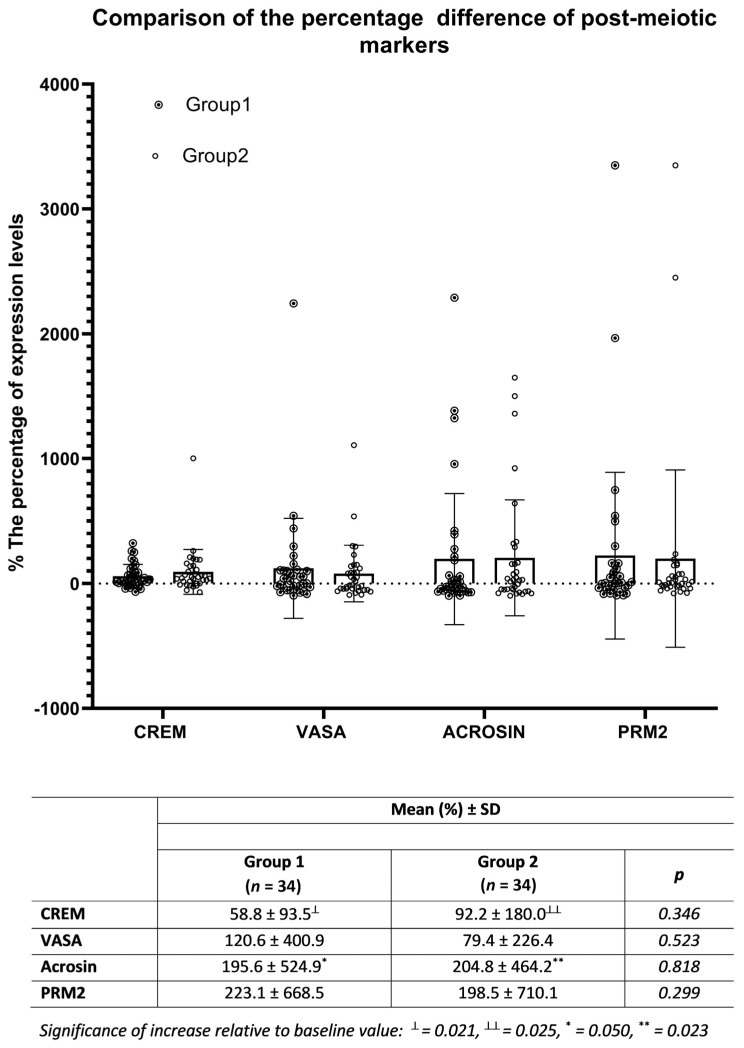
On day 7, comparison of the percent increases of CREM, VASA, acrosin, and PRM2 expression from pre-treatment values in cultures containing FSH/testosterone only (Group 1) with those of samples also co-cultured with healthy SCs (Group 2).

**Table 1 jcm-12-01073-t001:** Clinical parameters of the study groups.

	OA (*n* = 12)	NOA (*n* = 34)	*p*
Age (years)	31 ± 4.1	33 ± 2.6	n.s. *
Total testis volume (cc)	36 ± 6	18 ± 14	<0.05
FSH (IU/L)	3.6 ± 1.2	18.2 ± 9.3	<0.05
Total testosterone (ng/mL)	4.2 ± 1.2	2.6 ± 1.6	n.s. *

OA: obstructive azoospermia, NOA: non-obstructive azoospermia, * n.s. non-significant.

## Data Availability

The data presented in this study are available upon request from the corresponding author. The data are not publicly available because they contain information that can compromise the privacy of the research participants.

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
