# Peer review of "Co-Culture of Cryopreserved Healthy Sertoli Cells with Testicular Tissue of Non-Obstructive Azoospermia (NOA) Patients in Culture Media Containing Follicle-Stimulating Hormone (FSH)/Testosterone Has No Advantage in Germ Cell Maturation"

_jcm, 2023, doi:10.3390/jcm12031073_

Round 1

Reviewer 1 Report

 This study was conducted on the basis of clinical samples, so the conclusion is quite meaningful, but some raised points need to be addressed before publication. The comments as follows:

1. Fig1 Flow cytometry Figures are not clear enough. Please use figures not less than 300dpi.

2. What do P1,P2,P3,P4,P5,P6 refer to in Fig1A? Is it the number of cell passages? No explanation of this is found in either the method materials or the results.

3. It is mentioned in the results that "underwent apoptosis before freezing and after thawing were 16.1% and 15.2%, respectively" lines 288 and 289, Where did 16.1% and 15.2% come from? Is it the average of early apoptosis or the average of late apoptosis? Please explain in the results, and please comment the meanings of different regions in the streaming results.

4. If the apoptosis of cells before and after freeze-thaw is to be compared, the results of these two should be compared together and should not be divided into Fig1A and B. In addition, the parameters (voltage or restricted region) of A and B in Fig1 are obviously inconsistent, so the two cannot be compared. We deeply doubt your experimental conclusions.

5. Fig1-C represents the identification of SC. GATA4 and Vimentin proteins were mentioned in the manuscript as two key marker proteins of SC, but I only saw the immunofluorescence results of GATA4, not Vimentin.

In addition, you did not mark what green fluorescence represents in Fig1-C, but explained it in the figure notes. Please mark FITC directly in the figure. The results only include FITC and white light results, not merge results. In addition, SC cell positive controls are required, which will make the experimental results more rigorous.

6. Negative and positive controls should be added to Fig1D flow results

7. In Fig2, the increment of SC cells before and after freeze-thaw was detected. You introduced 6 times of repeated detection in the experimental method, and the average value was taken, which was only 6 times of detection in one experiment. However, independent repeated experiment is very important, and there is no independent repeated several times in the whole manuscript, please explain in the experimental method or the drawing notes.The size of the error bar should be reflected in the bar chart, although you describe it in detail in Table 2

8. In Fig2, the ordinate represents the absorption value at 550-690nm, but the absorption value is very low, less than 0.1. Did you subtract the blank control? Please describe this in the method materials.

9. The results of MTT experiment show that the absorption value of 550-690nm is too small, and the SD value is larger than Mean. Please give a reasonable explanation.

10. The immunohistochemical method was introduced in 2.1.2.2 of methods and materials, but no immunohistochemical results were seen. So you're talking about immunofluorescence here?

11. In Table 3 and 4, flow cytometry experiments were performed on the co-cultured cells, but they were only shown in the form of tables. The flow cytometry histogram should be added here to make the results more clear.

Reviewer 2 Report

The authors have undertaken a study to examine if Co-culture of cryopreserved healthy Sertoli cells with testicular tissue of non-obstructive azoospermia (NOA) patients in culture media containing follicle stimulating hormone (FSH)/testosterone has any advantage in germ cell maturation. The study appears to be well conducted but it needs minor revision.

The introduction is poorly written and the reference review is not well conducted. Please rewrite it.

Freeze protocol is not clear and should be revised.

The reference of the freezing protocol should be stated.

How many cells were measured by flow cytometry?

Flow cytometry graphs should be added to the text of the article.

Standard error should be added to the diagrams and the diagrams should be drawn with better quality.

Reviewer 3 Report

In general it is a very interesting article that deserves to be published. However, I make some suggestions for changes to improve it.

Introduction

Line 49, these percentages of incidence of what years are they? That is, has it been increasing? Is it current?

Line 58, why is the birth rate lower?

Line 60, what are those different techniques?

Materials and methods

Line 101, how NOA is diagnosed

In the methodology use min instead of minutes

The cryopreservation method is not mentioned in detail, it is only indicated in line 162. Need to be expanded 

Results

The results in the table are interesting, what is the reason for this increase in FSH and testicular volume in patients with NOA.

Why are testosterone levels not altered?

Discussion

This section needs to explain more about the results obtained. Line 367 points to cryopreservation as an important factor; however, this procedure is not described. Some references from the discussion could be updated. 

Conclusion

The use of SCs was not advantageous, it seems to me that a conclusion could be drafted in such a way that allows with the results obtained to know what could improve the maturation of testicular germ cells

Round 2

Reviewer 1 Report

The authors have answered the reviewers' comments.